# A Simple Thermodynamic Model of the Internal Convective Zone of the Earth

**DOI:** 10.3390/e20120985

**Published:** 2018-12-18

**Authors:** Karen Arango-Reyes, Marco Antonio Barranco-Jiménez, Gonzalo Ares de Parga-Álvarez, Fernando Angulo-Brown

**Affiliations:** 1Escuela Superior de Física y Matemáticas del Instituto Politécnico Nacional, Edif. 9 U.P. Zacatenco, Ciudad de México 07738, Mexico; 2Escuela Superior de Cómputo del Instituto Politécnico Nacional, Av. Miguel Bernard, Esq. Av. Miguel Othón de Mendizabal, Colonia Lindavista, Ciudad de México 07738, Mexico

**Keywords:** convective zone, earth’s mantle, finite time thermodynamics

## Abstract

As it is well known both atmospheric and mantle convection are very complex phenomena. The dynamical description of these processes is a very difficult task involving complicated 2-D or 3-D mathematical models. However, a first approximation to these phenomena can be by means of simplified thermodynamic models where the restriction imposed by the laws of thermodynamics play an important role. An example of this approach is the model proposed by Gordon and Zarmi in 1989 to emulate the convective cells of the atmospheric air by using finite-time thermodynamics (FTT). In the present article we use the FTT Gordon-Zarmi model to coarsely describe the convection in the Earth’s mantle. Our results permit the existence of two layers of convective cells along the mantle. Besides the model reasonably reproduce the temperatures of the main discontinuities in the mantle, such as the 410 km-discontinuity, the Repetti transition zone and the so-called D-Layer.

## 1. Introduction

The Earth’s mantle occupies more of the eighty percent of the total Earth volume. It lies between the core and the Earth’s crust (just before the Litosphere) [1,2]. The knowledge of the Earth’s mantle arises from the study of some volcanic activity waste, but mostly from experimental data provided by seismic waves. It is divided into two basic zones which are the inferior mantle (Mesosphere) and the superior mantle (Astenosphere). The Mesosphere extends from a depth of approximately 660 km up to the core-mantle limit and the Astenosphere continuous from the Mesosphere up to the Litosphere [3].

There are additional subdivisions: first, at the core-mantle boundary there is a very important region representing a transition known as Wiechert-Gutenberg discontinuity which is a change of composition known as “D-layer”; second, the zone of transition between the inferior mantle and the superior mantle is the discontinuity of Repetti; third, the 410 km-discontinuity; finally, in the limit of the mantle-crust, another zone of transition is the discontinuity of Mohorovicic (Moho). These discontinuities have been found due to several studies about the velocity changes of the seismic waves [1,2,4].

Within the Earth’s mantle there is a gradual increase in temperature with depth generating gradients that permit an efficient transmission of heat. However, it is well known that rocks are not good conductors of heat, therefore it can be concluded that the best way in which the mantle can transfer heat is through convection. The convection flow is the most important process that takes place inside the Earth, since it is the force that moves rigid lithospheric plates across the planet generating the formation of mountains, mountain ranges, volcanic and seismic activity of the Earth [1,2,3,4]. The first approaches to mantle convection were through 2-D models. However, nowadays the mantle convection is also modeled by means of 3-D models [5].

One of the most argued theories is the one of two layers of convection due to the seismic discontinuity of Repetti which is approximately at 660 km of depth, [5]. The existence of matter transport through this discontinuity was verified with the development of the seismic tomography, [5]. However, it is known that the transition zone marks a change in the convection of the mantle, which makes possible to think that at some early stage of Earth evolution there was a convection in two layers due to convection forces of the Earth [3,4,6].

One interesting application of finite-time thermodynamics (FTT) was made by Gordon and Zarmi (GZ) [7] taking the Sun-Earth-wind system as a FTT-cyclic heat engine where the heat input is solar radiation, the working fluid is the Earth’s atmosphere and the energy in the winds is the work per unit time produced. Later the GZ model for convective cells of the air in the atmosphere was extended by several authors [8,9,10,11,12,13] by taking into account some additional restrictions of the atmospheric behavior, such as the greenhouse effect, internal dissipation in the working fluid and some changes in the temperature of the cold thermal bath of the GZ model. In 2006 Reis and Bejan [14] also studied the natural convection loops in the atmospheric air by using the constructal theory [15]. Later, a connection of this theory with entropy generation was proposed [16]. Our approach is based on the fact that there are gradients of temperature within the Earth’s mantle; that is, it is necessary at least two representative mantle’s temperatures for making the creation of work possible; that is, to take the viscoelastic material of the mantle as a working fluid that converts heat into mechanical work. This permits to introduce in a natural way the concept of Earth mantle "heat engine". In this context, process variables as work rate, heat fluxes and efficiency for instance, find a simple theoretical framework, where thermodynamical restrictions play a major role. This is in contrast with disciplines as non-equilibrium thermodynamics and hydrodynamics based on local differential equations where the transition from local to global variables is not a trivial task, [17,18]. For the case of the GZ model for the convective cells of air within the Earth’s atmosphere these authors [7] found reasonable values for the annual average power in the Earth winds and for the average maximum and minimum temperatures of the atmosphere, without resorting to detailed dynamic models of the Earth’s atmosphere, and without considering any other effect; such as the Earth rotation, Earth translation around the Sun and ocean currents. These results by Gordon and Zarmi and those later found by several authors [8,9,10,11,12,13] by means of small variations of the GZ model indicate that the global thermodynamic approach to this kind of energy converters are good first approximations based only in the global restrictions imposed by the laws of thermodynamics and with the inclusion of the time through the methods of finite time thermodynamics [19,20,21,22,23,24,25]. We think that an analogous approach can be used for a first approximation to the convection zone of the Earth. In 2012, in Reference [26] the GZ model was also applied to the convective zone of the Sun. In the present article we apply the GZ model to the convective zone of the Earth, giving an additional step to the model of equilibrium thermodynamics presented by Stacey [27] for the mantle convection.

In this work, we apply the Gordon-Zarmi convective model to the convection in the Earth’s mantle, in a similar way that these authors used it firstly for convective atmospheric cells. The paper is organized as follows: For reasons of self-containment we include in Section 2 a brief review of finite-time thermodynamics and we also present a review of the GZ model applied to the convective atmospheric cells by means of two performance regimes (maximum power regime and maximum ecological regime). In Section 3 we apply the GZ model to the convective zone of the Earth’s mantle which is located between the core and the Earth’s crust and finally we present our conclusions.

## 2. Finite-Time Thermodynamics and the Gordon-Zarmi Convective Model

Since the pioneering Curzon-Ahlborn (CA) paper [28] published in 1975, FTT has been applied to several physical systems, as thermal engine models [20,21,22,23,24,29]. In the same way that early classical thermodynamics in the 19th century, starting from the study of thermal engines, soon reached practically all macroscopic systems, FTT embraced many problems where entropy production of global processes plays an unavoidable role. In a typical FTT heat engine model the whole entropy production is ascribed only to the coupling between the working substance and its surroundings and it is permitted that the working fluid undergoes only reversible transformations. This approach is called the endoreversibility hypothesis (EH) [30]. By means of this hyphotesis it has been possible to place realistic bounds on irreversible processes that proceed in finite time. Usually, in FTT methodology one calculates an extremum or optimum of a thermodynamically meaningful variable or functional [7]. In 2004, Fischer and Hoffmann [31] showed that a simple endoreversible model (the so-called Novikov engine) can reproduce the complex engine behavior of a quantitative dynamical simulation of an Otto engine including, but no limited to, effects from losses due to heat conduction, exhaust losses and frictional losses. On the other hand, Curto-Risso et al. [32] have published an FTT-model also for an irreversible Otto cycle for reproducing performance results of a very elaborated dynamical model of a real spark ignition heat engine including a turbulent flame propagation process, valves overlapping, heat transfer across the cylindrical walls, and a detailed analysis of the involved chemical reactions [33]. In References [31,32], the spirit of FTT is illustrated emphasizing the virtues and limitations of this methodology. However, in these articles the usefulness of FTT models is shown beyond any doubt. In fact, we can assert that the FTT spirit is concomitant with the spirit of a Carnotian thermodynamics in the sense of the search for certain kind of limits for thermodynamic variables and functionals. For example, in 1975, Curzon and Ahlborn [28] published an article where they proposed a kind of Carnot cycle that produces entropy only due to an irreversible Newtonian heat transfer between two thermal reservoirs at absolute temperatures $T_{1}$ and $T_{2}$
$\left(T_{1}>T_{2}\right)$ and the two isothermal branches of the working fluid at temperatures $T_{1w}$ and $T_{2w}$
$\left(T_{1w}>T_{2w}\right)$ respectively (see Figure 1).

CA used as heat transfer law the following linear equations,
(1)$$\begin{array}{ccc} Q_{1} & = & \alpha (T_{1}-T_{1w}), \end{array}$$
(2)$$\begin{array}{ccc} Q_{2} & = & \beta (T_{2w}-T_{2}), \end{array}$$
where α and β are the thermal conductances of the materials that separate the reservoirs from the working substance, and $Q_{1}$ and $Q_{2}$ are the heat flows per unit time. In this way, CA proposed an irreversible global model with $\Delta S_{tot}>0$ but internally reversible (EH). By integrating Equations (Equation 1) and (Equation 2), CA obtained the cycle’s period Δt and therefore they had a cycle with non-zero power, in contrast to the reversible Carnot cycle with both zero entropy and power productions. For the mentioned cycle model, CA maximized the power output, and they found that the efficiency under maximum-power conditions is expressed by,
(3)$$\eta_{CA}=1-\sqrt{\frac{T_{2}}{T_{1}}}.$$

This same expression was previously obtained by Chambadal [34] and Novikov [35]. Since the CA-paper, many works have been published in the FTT field and also different regimes of performance have been widely studied within the literature in order to characterize the energetic functions of endoreversible engines [7,20,21,22,23,24,25,28,29,30,31,32,36,37,38,39]. One regime of performance proposed by Angulo Brown [36] consists of finding the best trade-off between high power output and low entropy production, this regime is known as the ecological criterion.

In Figure 2, it can be observed the qualitative behavior of both the power *P* and the entropy production Σ in terms of the efficiency η, for the Curzon-Ahlborn cycle [20]. From this figure it can be seen that the ecological function,
(4)$$E=P-T_{2}\Sigma ,$$
is a concave curve with a maximum point. This function has the following properties: at maximum *E*, the power output satisfies $P_{E_{max}}\approx \frac{3}{4}P_{max}$; the entropy production is $\Sigma_{E_{max}}\approx \frac{1}{4}\Sigma_{P_{max}}$ and the efficiency is $\eta_{E_{max}}\approx \frac{1}{2}\left(\eta_{C}+\eta_{CA}\right)$ [40,41]. Due to the previous properties, the function given by Equation (Equation 4) was named the ecological function. The so-called ecological optimization has been applied in many areas. For instance, thermal engines [37], chemical engines [42,43,44], biochemical reactions [38,39], linear energy converters [38,39], superconducting transition [45], thermoeconomical optimization [46,47] and atmospheric convective cells [11,12,13].

The endoreversible model of Gordon and Zarmi [7] consists of a Curzon-Ahlborn cycle in the endoreversible limit and with instantaneous adiabats formed by four branches: (a) Two isothermal branches, one in which the atmosphere absorbs solar radiation at low altitudes and one in which the atmosphere rejects heat at high altitudes to the universe; and (b) two intermediate instantaneous adiabats with rising and falling air currents (see Figure 3). According to these authors this oversimplified Carnot-like heat engine corresponds very approximately to the global scale motions of winds in convective cells.

### Atmospheric Convection

The endoreversible model of Gordon-Zarmi (GZ) is based on annual average quantities in such a way that it represents only 2-dimensional virtual convective cells. On the other hand, quantities such as the work done by the working fluid, the internal energy of the fluid and the annual average flux of solar radiation are expressed in units of the earth’s surface.

$T_{1}$ is the temperature that the working fluid has in an isotherm at a low altitude, in this first half of the cycle, the working fluid absorbs solar radiation arising up adiabatically to the isotherm $T_{2}$ at high altitudes, and in this case the working fluid rejects heat in the form of black body radiation to the cold reservoir at temperature $T_{ext}$. In the GZ model, the objective is to maximize the work done per cycle, subject to the endoreversibility condition; that is,
(5)$$\Delta S_{int}=\int_{0}^{t_{0}}\left\{\frac{q_{S}\left(t\right)-\sigma \left[T^{4}\left(t\right)-T_{ext}^{4}\left(t\right)\right]}{T(t)}\right\}dt=0,$$
where $\Delta S_{int}$ is the entropy change per unit area along the cycle, $t_{0}$ is the cycle duration time, $T_{ext}=3$ K (Surrounding Universe), σ is the Stefan-Boltzmann constant, and finally for $q_{s}\left(t\right)$ and $T\left(t\right)$, the following definitions are used [7],
(6)$$\begin{array}{cc} T\left(t\right)= & \{T_{1}\;\mathrm{for}\;0\leq t\leq \frac{t_{0}}{2}\;\mathrm{and}\;T_{2}\;\mathrm{for}\;\frac{t_{0}}{2}\leq t\leq t_{0},\\ q_{S}\left(t\right)= & \{\frac{Isc\left(1-\rho \right)}{2}\;\mathrm{for}\;0\leq t\leq \frac{t_{0}}{2}\;\mathrm{and}\;0\;\mathrm{for}\;\frac{t_{0}}{2}\leq t\leq t_{0}, \end{array}$$
being $I_{sc}=1373$ W/m${}^{2}$ (yearly average solar constant) and ρ= 0.35 is the effective average albedo of the Earth’s atmosphere, being this, the only macroscopic quantity characterizing the Earth’s atmosphere [7]. In Equation (Equation 6) the temporal dependence of the temperatures T(t) and the heat input $q_{s}\left(t\right)$ is given in terms of the lower part and the upper part of the cycle period $t_{0}$ respectively. The work per cycle *W* is taken from the first law of thermodynamics,
(7)$$\Delta U=-W+\int_{0}^{t_{0}}\left\{q_{S}\left(t\right)-\sigma \left[T^{4}\left(t\right)-T_{ext}^{4}\left(t\right)\right]\right\}dt.$$

On the other hand, the following averages are defined,
(8)$$\begin{array}{ccc} \bar{T} & = & \frac{T_{1}+T_{2}}{2},\\ \bar{T^{n}} & = & \frac{T_{1}^{n}+T_{2}^{n}}{2},\\ \bar{q_{s}} & = & \frac{Isc\left(1-\rho \right)}{4}, \end{array}$$
where *n* = 3 or 4, and the factor $\frac{1}{4}$ of the average of $q_{s}$ appears when considering $\frac{1}{2}$ due to taking into account the day and night, and another $\frac{1}{2}$ is a geometric factor that has to do with the Earth’s cross section. With the previous considerations, GZ proposed a Lagrangian for maximizing the work per unit time that is calculated from Equations (Equation 7) and (Equation 8), and including the endoreversibility constraint given by Equation (Equation 5), thus the Lagrangian results [7],
(9)$$L=T^{4}\left(t\right)+\lambda \left[\frac{q_{s}\left(t\right)}{T\left(t\right)}-\sigma T^{3}\left(t\right)\right],$$
where λ is the Lagrange multiplier. Through the Euler-Lagrange formalism, Gordon-Zarmi found the following values for the Earth’s atmosphere, $T_{1}=277$ K, $T_{2}=192$ K, and $P_{max}$ = $W_{max}$ / $t_{0}$ = 17.1 W/m${}^{2}$. The values for $P_{max}$ and $T_{1}$, are not so far from actual values, which are $P_{max}=7$ W/m${}^{2}$ [48] and $T_{1}=290$ K at ground level. However, the value for $T_{2}=192$ K corresponds to temperatures that are found in the air at altitudes of around 75–90 km. As it is well known, the convective motions of air occurs mainly within the troposphere.

In Reference [11] another endoreversible case was analyzed, but using as a cold reservoir the well known tropopause shell with $T_{ext}=200$ K, for this case the proposed Lagrangian was [11],
(10)$$L=\bar{q_{s}+\sigma T_{ext}^{4}}-\sigma \bar{T^{4}}-\gamma \left[\frac{\bar{q_{s}}}{T_{1}}-\sigma \frac{T_{1}^{3}+T_{2}^{3}}{2}-\sigma T_{ext}^{4}\left(\frac{1}{T_{1}}+\frac{1}{T_{2}}\right)\right],$$
with γ the Lagrange multiplier, and by numerically solving $\partial L/\partial T_{1}=0$ and $\partial L/\partial T_{2}=0$, $T_{1}=293.387$ K and $T_{2}=239.267$ K were obtained, which are good values for the convective cells restricted to the troposphere, since in that zone mainly occurs the activity of the terrestrial winds. To calculate the wind power they used [11],
(11)$$P=\bar{q_{s}+\sigma T_{ext}^{4}}-\sigma \bar{T^{4}},$$
obtaining *P* = 10.758 W/m${}^{2}$, which is also a good value for the wind power [48]. In the same Reference [11], the maximum ecological criterion also was applied to the GZ model, which consists in maximizing the Equation (Equation 4).

By means of the second law of thermodynamics the authors calculated the total entropy change per cycle,
(12)$$\Delta S_{U}=\int_{0}^{t_{0}}\left\{\frac{-q_{S}\left(t\right)+\sigma \left[T^{4}\left(t\right)-T_{ext}^{4}\left(t\right)\right]}{T\left(t\right)}\right\}dt.$$

By using Equation (Equation 8),
(13)$$\Delta S_{U}=\int_{0}^{\frac{t_{0}}{2}}\left\{\frac{-q_{S}\left(t\right)}{T_{1}}+\sigma \left[T_{1}^{3}-\frac{T_{ext}^{4}}{T_{1}}\right]\right\}dt-\int_{\frac{t_{0}}{2}}^{t_{0}}\left\{\frac{\sigma \left[T_{2}^{4}-T_{ext}^{4}\right]}{T_{ext}}\right\}dt.$$

The total entropy production is given by [12],
(14)$$\Sigma =\frac{\Delta S_{U}}{t_{0}}\approx \frac{\bar{q_{s}}}{T_{1}}+\frac{\sigma }{2}\left(T_{1}^{3}+\frac{T_{2}^{4}}{T_{ext}}\right).$$

By the sustitution of Equations (Equation 11) and (Equation 14) into Equation (Equation 4), the ecological function is,
(15)$$E=\bar{q_{s}+\sigma T_{ext}^{4}}-\sigma \bar{T^{4}}+\frac{T_{ext}}{T_{1}}\left(\bar{q_{s}}+\frac{\sigma T_{ext}^{4}}{2}\right)-\frac{\sigma T_{ext}}{2}\left(T_{1}^{3}+\frac{T_{2}^{4}}{T_{ext}}\right)-\frac{\sigma T_{ext}^{4}}{2},$$
for the case where $T_{ext}=3$ K, the authors took into account the approximation $\bar{q_{s}}\gg$
$\sigma T_{ext}^{4}$ (223 W/m${}^{2}$≫ 4.59 × 10${}^{-6}$ W/m${}^{2}$), and they proposed the following Lagrangian [11],
(16)$$L_{E}=\bar{q_{s}}-\sigma \bar{T^{4}}+\frac{T_{ext}\bar{q_{s}}}{T_{1}}-\frac{\sigma T_{ext}}{2}\left(T_{1}^{3}+\frac{T_{2}^{4}}{T_{ext}}\right)-\gamma \left[\frac{\bar{q_{s}}}{T_{1}}-\sigma \bar{T^{3}}\right],$$
with which they found the following values, $T_{1}$ = 294.08 K, $T_{2}$ = 109.54 K and *P* = 6.89 W/m${}^{2}$, which are reasonable values for $T_{1}$ and *P* but not for $T_{2}$. Repeating the calculations, now with $T_{ext}$ = 200 K, they propose a new Lagrangian [11],
(17)$$\begin{array}{cc} L_{E}= & \bar{q_{s}+\sigma T_{ext}^{4}}-\sigma \bar{T^{4}}+\frac{T_{ext}}{T_{1}}\left(\bar{q_{s}}+\frac{\sigma T_{ext}^{4}}{2}\right)-\frac{\sigma T_{ext}}{2}\left(T_{1}^{3}+\frac{T_{2}^{4}}{T_{ext}}\right)\\ & -\frac{\sigma T_{ext}^{4}}{2}-\gamma \left[\frac{\bar{q_{s}}}{T_{1}}-\sigma \bar{T^{3}}+\sigma T_{ext}^{4}\left(\frac{1}{T_{1}}+\frac{1}{T_{2}}\right)\right], \end{array}$$

Obtaining as results $T_{1}=303$ K, $T_{2}=219$ K and *P* = 7 W/m${}^{2}$, which finally are very good values compared with the current ones [48]. Obviously, the actual rising and falling air currents in atmospheric convective cells are not instantaneous. However, the numerical results obtained with the GZ-model are not so far from actual values. Thus, we can consider that the hypothesis of instantaneous adiabats is reasonable; that is, the adiabatic times are much smaller than the isothermic times. This last fact was corroborated by Agrawal et al. [49] for the case of typical finite time heat engine models.

## 3. GZ Model Applied to the Convective Zone of the Earth

The temperature distribution throughout the Earth’s mantle has been studied by several authors [1,2,5]. The main mechanism of heat transport along the mantle is by means of convective motions [1,2,5,50]. As it is well known solid-state convection in Earth’s mantle may be approximated as a fluid dynamical process involving many complicated physical effects, such as brittle failure in the surface boundary layers (plates), interior rheology which may depend strongly upon temperature, pressure and shear stress, both endothermic and exothermic mineralogical phase changes, both internal heating from radiactivity and bottom heating from the core, and possible chemical stratification [5,50]. According to Bunge et al. [5], the mantle convects with an effective Rayleigh number on the order of 10${}^{8}$ to 10${}^{9}$. Typical mantle convection speed is around 20 mm/year. A single shallow convection cycle takes on the order of 50 Myr, though deeper convection can be closer to 200 Myr [5,50]. The question if the mantle’s convection is layered or it behaves as a whole cycle is an open debate yet [5,50]. In the present section we propose a simplified 2-D model for the mantle’s convection following an analogous approach to that used by Gordon and Zarmi to emulate the convective motion of the air in the Earth atmosphere without resorting to detailed dynamic models of the atmosphere, and without considering any other effect; such as the Earth rotation, Earth translation around the Sun and ocean currents. In our proposed model for mantle convection we overlook the complicated physical effects above mentioned. Our model takes as known data the temperature of the core-mantle interface ($T_{1}$) and the temperature of the interface of the lithosphere with the upper mantle ($T_{2}$), being these temperatures the only data characterizing the mantle properties. In this way, the quantities to be determined are the so-called working temperatures ($T_{1w}$ and $T_{2w}$), corresponding to the isotherms of an internal Carnot-type cycle carried out by the working substance, which in this case is the viscoelastic material of the mantle; that is, here we are using the typical FTT terminology for a Curzon-Ahlborn cycle (see Figure 1). The flow of heat that drives the cycle is that coming from the core of the Earth, which we model using a heat transfer law of the Dulong-Petit type [45,51], which is used as an approximation to describe combined conductive-convective and radiative cooling by a power-law relationship of the form,
(18)$$\frac{dQ}{dt}=\alpha {\left(T_{a}-T\right)}^{n},$$
where dQ/dt is the rate of heat loss per unit area from a body at temperature *T*, α is a thermal conductance (or a coefficient of convection), $T_{a}$ is the temperature of the fluid surrounding the body and *n* is an exponent with value between 1.1 and 1.6 [20]. Some authors have established that n=5/4 based on studies made by Lorentz and Langmuir [20]. As O’Sullivan asserts [51], Stefan in his original 1879 paper, took the results of Dulong and Petit (DP) along with experiments by Tyndall and pointed out that the DP model was in agreement with his $T^{4}$ law [51].

In Section 3.1 we apply the GZ model to the convective zone of the Earth assuming a maximum power performance regime. In Section 3.2 we study the GZ convective model under maximum ecological function regime.

### 3.1. Maximum Power Regime

According to some authors [1,52] the geothermal gradient estimated for the Earth interior and the temperature values assigned to the different interior boundaries can vary in approximately 500 ${}^{{}^{\circ}}$C [1,2,52]. The value of temperatures from the inner core until the superior mantle varies approximately from 4500 ${}^{{}^{\circ}}$C until 1200–1500 ${}^{{}^{\circ}}$C (below Litosphere) [1,2]. In Figure 4, we show the heat fluxes balance within the convective zone of Earth’s mantle. In this section, we apply the GZ model to the convective zone of the Earth, in this case for our thermodynamic analysis we use the law of Dulong-Petit to describe the heat transfer [45].

By applying the endoreversibility condition to the fluxes shown in Figure 4. we have,
(19)$$\Delta S_{int}=\int_{0}^{t_{0}}\frac{\alpha {\left(T_{1}-T_{1w}\right)}^{\frac{5}{4}}-\alpha {\left(T_{2w}-T_{2}\right)}^{\frac{5}{4}}}{T\left(t\right)}dt=0,$$
here, in Equation (Equation 18) we use the same coefficient of convection (α) for both the lower and upper parts of the Carnot-like cycle emulating a convective cell, under the assumption that in average the working substance (mantle materials) undergoing that convection movement has practically the same properties. By using the following definitions,
(20)$$\begin{array}{c} T\left(t\right)=\{T_{1w}\;\mathrm{for}\;0\leq t\leq \frac{t_{0}}{2}\;\mathrm{and}\;T_{2w}\;\mathrm{for}\;\frac{t_{0}}{2}\leq t\leq t_{0}, \end{array}$$
which give us the temporal dependence of the working temperatures along the two isotherms (lower and upper) of the cycle. By Using Equations (Equation 19) and (Equation 20) we obtain,
(21)$$\frac{{\left(T_{1}-T_{1w}\right)}^{\frac{5}{4}}}{T_{1w}}-\frac{{\left(T_{2w}-T_{2}\right)}^{\frac{5}{4}}}{T_{2w}}=0.$$

On the other hand, from the first law of thermodynamics, ΔU=Q−W=0, we obtain,
(22)$$\begin{array}{ccc} W & = & Q=\int {}_{0}^{t_{0}}\left[\alpha {\left(T_{1}-T_{1w}\right)}^{\frac{5}{4}}-\alpha {\left(T_{2w}-T_{2}\right)}^{\frac{5}{4}}\right]dt\\ & = & \left[\alpha {\left(T_{1}-T_{1w}\right)}^{\frac{5}{4}}-\alpha {\left(T_{2w}-T_{2}\right)}^{\frac{5}{4}}\right]t_{0}, \end{array}$$
that is, the power output ($P=W/t_{0}$) results,
(23)$$P=\alpha {\left(T_{1}-T_{1w}\right)}^{\frac{5}{4}}-\alpha {\left(T_{2w}-T_{2}\right)}^{\frac{5}{4}}.$$
with Equations (Equation 21) and (Equation 23) we propose the following Lagrangian,
(24)$$L=\alpha {\left(T_{1}-T_{1w}\right)}^{\frac{5}{4}}-\alpha {\left(T_{2w}-T_{2}\right)}^{\frac{5}{4}}-\lambda \left[\frac{{\left(T_{1}-T_{1w}\right)}^{\frac{5}{4}}}{T_{1w}}-\frac{{\left(T_{2w}-T_{2}\right)}^{\frac{5}{4}}}{T_{2w}}\right],$$
where λ is a Lagrange multiplier. By means of $\partial L/\partial T_{1w}=0$ and $\partial L/\partial T_{2w}=0$, we get,
(25)$$-\lambda \left[-\frac{5}{4}\frac{{\left(T_{1}-T_{1w}\right)}^{\frac{1}{4}}}{T_{1w}}+\frac{{\left(T_{1}-T_{1w}\right)}^{\frac{5}{4}}}{T_{1w}^{2}}\right]=\frac{5\alpha }{4}{\left(T_{1}-T_{1w}\right)}^{\frac{1}{4}},$$
and
(26)$$-\lambda \left[-\frac{5}{4}\frac{{\left(T_{2w}-T_{2}\right)}^{\frac{1}{4}}}{T_{2w}}+\frac{{\left(T_{2w}-T_{2}\right)}^{\frac{5}{4}}}{T_{2w}^{2}}\right]=\frac{5\alpha }{4}{\left(T_{2w}-T_{2}\right)}^{\frac{1}{4}}.$$

From Equations (Equation 25) and (Equation 26) we have,
(27)$$-\frac{5}{4}\frac{1}{T_{1w}}-\frac{\left(T_{1}-T_{1w}\right)}{T_{1w}^{2}}+\frac{5}{4}\frac{1}{T_{2w}}-\frac{\left(T_{2w}-T_{2}\right)}{T_{2w}^{2}}=0.$$

By numerically solving Equations (Equation 21) and (Equation 27) we obtain $T_{1w}$ = 3517.36 ${}^{{}^{\circ}}$C and $T_{2w}$ = 2166.97 ${}^{{}^{\circ}}$C (all our calculations for mantle convection are made by using Kelvin temperatures, and then converted in Celsius ones). These first values for $T_{1w}$ and $T_{2w}$ are close to those reported values in the literature [1,2] for the well known transition zones of the Earth’s mantle named D layer and the Repetti transition zone respectively (see Figure 5).

By numerically solving again the system of Equations (Equation 21) and (Equation 27), but now by using $T_{1}$ = 2166.97 ${}^{{}^{\circ}}$C (that is, the value of $T_{2w}$ in the previous case) as the hot reservoir temperature of the second isothermal layer we find the following values, $T_{1w}^{\prime }$ = 1969.85 ${}^{{}^{\circ}}$C and $T_{2w}^{\prime }=1672.98$
${}^{{}^{\circ}}$C. These temperatures correspond approximately to the well known Repetti transition zone, that in the case of Figure 5 lies between 1969.85 °C and 2166.97 ${}^{{}^{\circ}}$C. Interestingly, besides the simplicity of the GZ-model, it leads to a reasonable stratification of the temperatures corresponding to the different layers of the Earth’s interior. In fact, a first application of the GZ-model between the temperature of the boundary of the inner core and the lower mantle and the temperature of the inferior crust results in a convective layer that corresponds very well with the lower mantle, in such a form that a second application of the GZ-model is possible producing a second convective layer between the Repetti layer and the lower part of the crust, perhaps including the Moho discontinuity. That is, the simple GZ-model of convection is compatible with the models that support a layered convection throughout the mantle (two layers in this case).

### 3.2. Ecological Function Regime

Now, our objective is the maximization of the so-called ecological function defined by Equation (Equation 4), instead of power output. First, we calculate $\Delta S_{U}$, the total entropy change per cycle (system plus surroundings), that is $\Delta S_{U}=\Delta S_{sys}+\Delta S_{surr}$, then from Figure 4 we get,
(28)$$\Delta S_{U}=\left[{\left(T_{1}-T_{1w}\right)}^{\frac{5}{4}}\left(\frac{1}{T_{1w}}-\frac{1}{T_{1}}\right)+{\left(T_{2w}-T_{2}\right)}^{\frac{5}{4}}\left(\frac{1}{T_{2}}-\frac{1}{T_{2w}}\right)\right]\frac{\alpha t_{0}}{2}.$$

Therefore, the entropy production is,
(29)$$\Sigma =\frac{\Delta S_{U}}{t_{0}}=\left[{\left(T_{1}-T_{1w}\right)}^{\frac{5}{4}}\left(\frac{1}{T_{1w}}-\frac{1}{T_{1}}\right)+{\left(T_{2w}-T_{2}\right)}^{\frac{5}{4}}\left(\frac{1}{T_{2}}-\frac{1}{T_{2w}}\right)\right]\frac{\alpha }{2}.$$

By substituting Equations (Equation 28) and (Equation 29) into Equation (Equation 4) the ecological function results,
(30)$$E=\left[{\left(T_{1}-T_{1w}\right)}^{\frac{5}{4}}\left(2-\frac{T_{2}}{T_{1w}}+\frac{T_{2}}{T_{1}}\right)-{\left(T_{2w}-T_{2}\right)}^{\frac{5}{4}}\left(3-\frac{T_{2}}{T_{2w}}\right)\right]\frac{\alpha }{2}.$$

With Equations (Equation 21) and (Equation 30) we propose the following Lagrangian,
(31)$$\begin{array}{cc} L_{E}= & \left[{\left(T_{1}-T_{1w}\right)}^{\frac{5}{4}}\left(2-\frac{T_{2}}{T_{1w}}+\frac{T_{2}}{T_{1}}\right)-{\left(T_{2w}-T_{2}\right)}^{\frac{5}{4}}\left(3-\frac{T_{2}}{T_{2w}}\right)\right]\frac{\alpha }{2}\\ & -\lambda_{E}\left[\frac{{\left(T_{1}-T_{1w}\right)}^{\frac{5}{4}}}{T_{1w}}-\frac{{\left(T_{2w}-T_{2}\right)}^{\frac{5}{4}}}{T_{2w}}\right], \end{array}$$
being $\lambda_{E}$ a Lagrange multiplier. By means of the extremal conditions ($\partial L_{E}/\partial T\left(t\right)=0$), we obtain the following equations,
(32)$$\begin{array}{c} \lambda_{E}\left[-\frac{5}{4}\frac{{\left(T_{1}-T_{1w}\right)}^{\frac{1}{4}}}{T_{1w}}-\frac{{\left(T_{1}-T_{1w}\right)}^{\frac{5}{4}}}{T_{1w}^{2}}\right]\\ =\left[-\frac{5}{4}{\left(T_{1}-T_{1w}\right)}^{\frac{1}{4}}\left(2-\frac{T_{2}}{T_{1w}}+\frac{T_{2}}{T_{1}}\right)+\frac{T_{2}}{T_{1w}^{2}}{\left(T_{1}-T_{1w}\right)}^{\frac{5}{4}}\right]\frac{\alpha }{2}, \end{array}$$
and
(33)$$\begin{array}{c} \lambda_{E}\left[-\frac{5}{4}\frac{{\left(T_{2w}-T_{2}\right)}^{\frac{1}{4}}}{T_{2w}}+\frac{{\left(T_{2w}-T_{2}\right)}^{\frac{5}{4}}}{T_{2w}^{2}}\right]\\ =\left[-\frac{5}{4}{\left(T_{2w}-T_{2}\right)}^{\frac{1}{4}}\left(3-\frac{T_{2}}{T_{2w}}\right)-\frac{T_{2}}{T_{2w}^{2}}{\left(T_{2w}-T_{2}\right)}^{\frac{5}{4}}\right]\frac{\alpha }{2}. \end{array}$$

From the previous equations we obtain,
(34)$$\begin{array}{c} \left[-\frac{5}{4}\frac{{\left(T_{1}-T_{1w}\right)}^{\frac{1}{4}}}{T_{1w}}-\frac{{\left(T_{1}-T_{1w}\right)}^{\frac{5}{4}}}{T_{1w}^{2}}\right]\left[-\frac{5}{4}{\left(T_{2w}-T_{2}\right)}^{\frac{1}{4}}\left(3-\frac{T_{2}}{T_{2w}}\right)-\frac{T_{2}}{T_{2w}^{2}}{\left(T_{2w}-T_{2}\right)}^{\frac{5}{4}}\right]\\ -\left[-\frac{5}{4}\frac{{\left(T_{2w}-T_{2}\right)}^{\frac{1}{4}}}{T_{2w}}+\frac{{\left(T_{2w}-T_{2}\right)}^{\frac{5}{4}}}{T_{2w}^{2}}\right]\left[-\frac{5}{4}{\left(T_{1}-T_{1w}\right)}^{\frac{1}{4}}\left(2-\frac{T_{2}}{T_{1w}}+\frac{T_{2}}{T_{1}}\right)+\frac{T_{2}}{T_{1w}^{2}}{\left(T_{1}-T_{1w}\right)}^{\frac{5}{4}}\right]\\ =0 \end{array}$$

Now solving Equations (Equation 21) and (Equation 34) we obtain $T_{1w}$ = 3705.38 ${}^{{}^{\circ}}$C and $T_{2w}$ = 1981.63 ${}^{{}^{\circ}}$C. By numerically solving the system of Equations (Equation 21) and (Equation 34), but now using $T_{1}$ = 1981.63 ${}^{{}^{\circ}}$C; that is, the previous $T_{2w}$ is taken now as the hot bath for the second layer. This way we obtain, $T_{1w}=1882.12$
${}^{{}^{\circ}}$C and $T_{2w}$ = 1586.81 ${}^{{}^{\circ}}$C as values for the second convection layer (see Figure 6). Thus, for both optimization criteria (maximum power and maximum ecological function) the set of temperature values obtained shown in Figure 5 and Figure 6 are not so far from the accepted values for the temperatures of the boundaries between the different strata of the inner region between outer core and the crust.

In Table 1 and Table 2, we show the working temperatures ($T_{1w}$, $T_{2w}$, $T_{1w}^{\prime }$ and $T_{2w}^{\prime }$) for the two convective layers that fit in the available space between the outer core and the lower limit of the crust; that is, the place that the mantle occupies. Table 1 corresponds to the results stemming from the maximization of power output of the convective model and Table 2 are the corresponding results for the maximum ecological regime. The numerical results shown in both Tables were obtained by means of the GZ-model used in Section 3 for an exponent n=5/4. In these Tables the exponents *n* = 1, 1.2, 1.25 and 1.5 were used. All these exponents were reported as reasonable values for the Dulong-Petit law [51]. As we can see in both Tables all the calculated values for the working temperatures are no so far from the estimated values for the corresponding strata of the interior of the Earth located between the outer core and the lower part of the crust, especially if we consider that the values reported for the interior temperatures have an inaccuracy of around 500 ${}^{{}^{\circ}}$C [1,52]. For example, in the core-mantle boundary; that is, the D-layer, which has approximately a thickness between 200 and 250 km, the estimated value for the temperature gradient is around 3 ${}^{{}^{\circ}}$C/km and therefore, the temperature change along the D-layer is between 600 and 750 ${}^{{}^{\circ}}$C with the aforementioned inaccuracy of 500 ${}^{{}^{\circ}}$C. Thus, the temperature changes calculated with $T_{1}=4500\;{}^{{}^{\circ}}$C and $T_{1w}$ taken from both Tables are within the reported range for the D-layer [1,52]. As we said before the elapsing time of actual convective cells in the mantle is in the order of 50–200 Myr; that is, they are very slow processes. We think that for this kind of processes both the maximum power and the maximum ecological regimes are not very different.

## 4. Conclusions

As it is well known both atmospheric and mantle convections are very complex phenomena [53]. The dynamical description of these processes is a very difficult task involving complicated mathematical models. Some of the approaches to this problem are based on numerical 2D and 3D computational formulations [3,5,6]. However, a first approximation to these phenomena can be throughout simplified thermodynamic models where the restrictions imposed by the laws of thermodynamics play the main role [1,5]. For instance, Gordon and Zarmi in 1989 proposed a simplified model representing the convective cells of the atmospheric air without resorting to detailed dynamic models of the Earth’s atmosphere, and without considering any other effect, such as the Earth rotation, Earth translation around the Sun and ocean currents. In spite of the simplicity of that model, the obtained values for the annual average power of the winds and the mean temperature of the Earth surface were reasonable. Later, other authors by means of small changes on the GZ-model also obtained very good results for the mentioned quantities and also for the temperature of the high part of the troposphere [8,9,11,12]. In the present article we propose a thermodynamic simplified model for the convective zone of the Earth’s mantle by means of the GZ-model. Interestingly, also for the convective cells in the mantle the GZ-model leads to reasonable values for the temperatures of the boundaries between the different strata existing through the mantle. For example, in Figure 5 we observe that between the core-mantle boundary at $T_{1}=4500\;{}^{{}^{\circ}}$C and the mantle-litosphere boundary at $T_{2}=1500\;{}^{{}^{\circ}}$C a first layer of convective cells is generated by the model with working temperatures $T_{1w}=3517.36\;{}^{{}^{\circ}}$C and $T_{2w}=2166.97\;{}^{{}^{\circ}}$C. Curiously, the temperature interval between $T_{1}$ and $T_{1w}$ corresponds very well with the *D*-layer. On the other hand, between $T_{2w}$ and $T_{2}$, the GZ-model permits the existence of a second layer of convective cells. This second convective cell is obtained by taking $T_{2w}=2166.97\;{}^{{}^{\circ}}$C as the hot reservoir of a “heat engine” with a cold reservoir given by $T_{2}=1500\;{}^{{}^{\circ}}$C. In this way, we obtained two new working temperatures $T_{1w}^{\prime }=1969.85\;{}^{{}^{\circ}}$C and $T_{2w}^{\prime }=1672.98\;{}^{{}^{\circ}}$C (see Figure 5). Moreover, the temperatures interval between $T_{2w}$ and $T_{1w}^{\prime }$ corresponds also very well with the Repetti discontinuity. Besides, the higher interval of temperatures in Figure 5 between $T_{2w}^{\prime }=1672.98\;{}^{{}^{\circ}}$C and $T_{2}=1500\;{}^{{}^{\circ}}$C is close to the so-called 410 km-discontinuity [1,5]. As we said in Section 4, the temperature values shown in Figure 5 were obtained with the GZ-model under a maximum power regime. In that section we also made a maximization of the GZ-model under maximum ecological function. In the third row of Table 2 we show the temperature values for the ecological regime for the case of n=5/4. As we can see, these temperatures also correspond approximately with the strata boundaries previously mentioned for the case of maximum power (see Table 1 third row). In Table 1, we also show the temperature values for other exponents of Equation (Equation 18) under maximum power regime and analogously in Table 2, we show the corresponding values for the ecological regime. In summary, we can observe in both Tables that for all the current values of the exponent *n* in Equation (Equation 18), the temperatures obtained are within the ranges corresponding to the mentioned strata boundaries. Remarkably, all the temperatures calculated with the GZ-model under the two optimization criteria used seemingly are independent of the coefficient α in Equation (Equation 18); that is, independent from the coefficient of convection. As we mentioned for the case of the atmospheric air convective cells the hypothesis of considering the adiabatic branches as instantaneous works well because the adiabatic times are much smaller than the isothermal times. In spite of the very large duration of the mantle convective cells (of the order of several tens of Myr), judging by our numerical results this hypothesis is also reasonable for mantle convection. It is convenient to emphazise that in the GZ-model for the air convective cells the input data are the solar constant, the temperature of the cold reservoir and the albedo parameter, being this the unique parameter depending on the global atmospheric features. On the other hand the unknowns of the model are the surface temperature of the Earth ($T_{1}$) taking as equal to the air temperature at low altitudes ($T_{1w}=T_{1}$) and the second unknown is $T_{2w}$; that is, the temperature of the air at high altitudes. In the case of GZ model applied to the mantle convection, we have taken as input data of the model the temperatures $T_{1}$ and $T_{2}$, being these temperatures the unique data that has to do with the general chemical and physical features of the mantle. The unknowns of the model are the working temperatures $T_{1w}$ and $T_{2w}$ which result in a good approximation coincident with the temperatures corresponding with the accepted values of the different transitions zones (the D-layer, the Repetti discontinuity and the 410 km-discontinuity). Finally, we wish to remark that nevertheless the simplicity of the GZ-model it is capable to offer a general view of this phenomenon compatible with the thermodynamics laws.

## Figures and Tables

**Figure 1 entropy-20-00985-f001:**
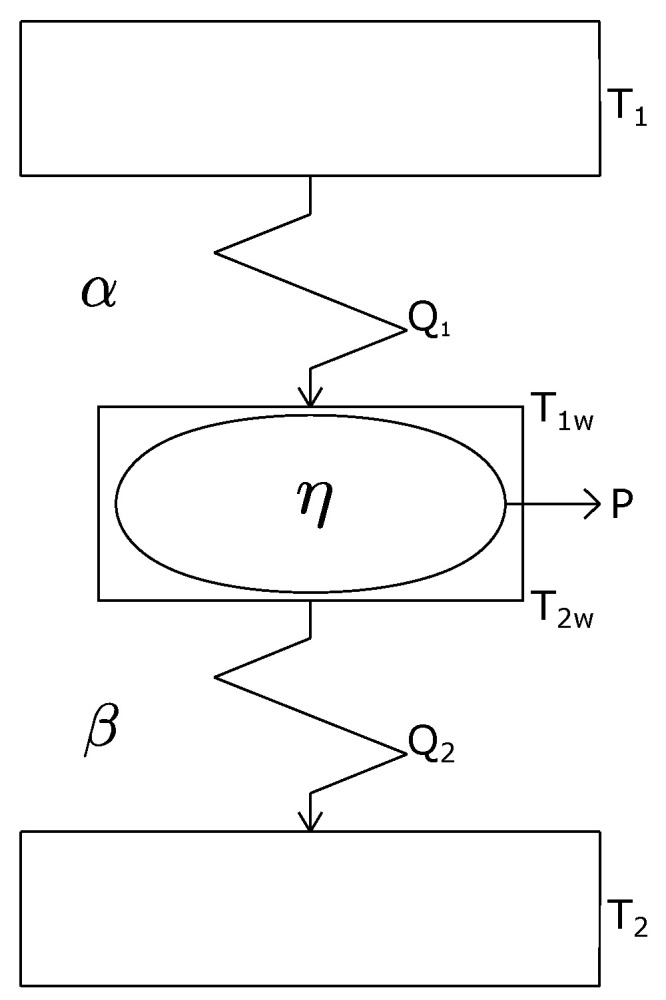
Curzon and Ahlborn heat engine model.

**Figure 2 entropy-20-00985-f002:**
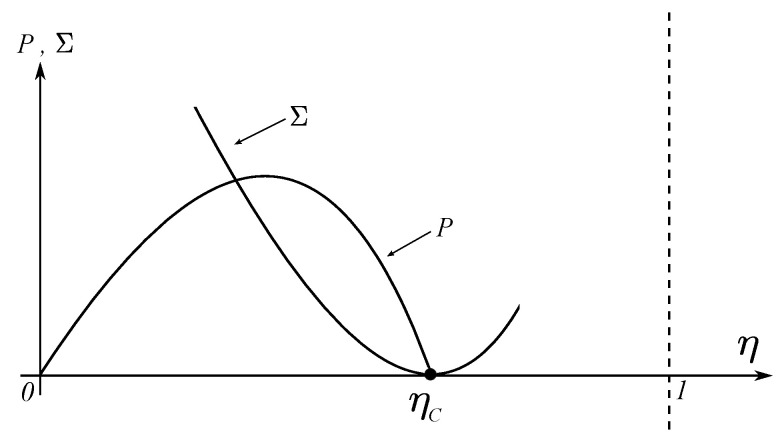
Behavior of power and entropy production vs internal efficiency.

**Figure 3 entropy-20-00985-f003:**
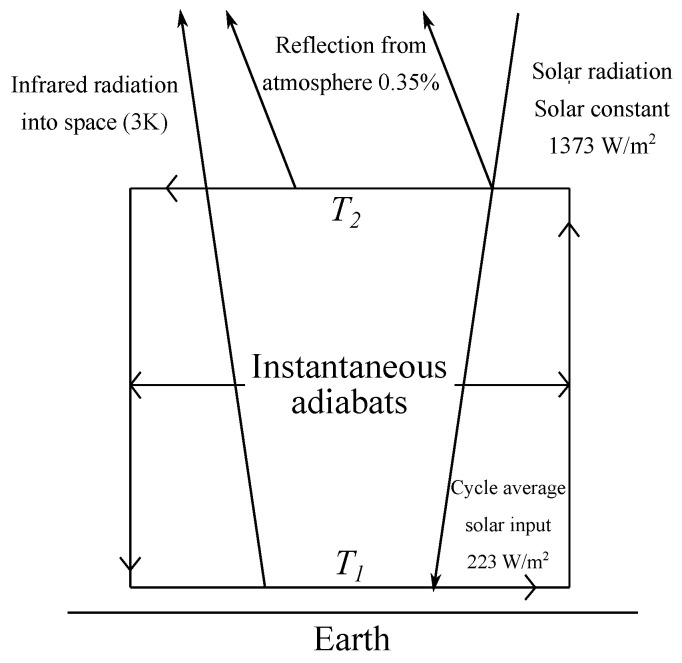
Scheme of a simplified solar-driven heat engine (taken from Reference [7]).

**Figure 4 entropy-20-00985-f004:**
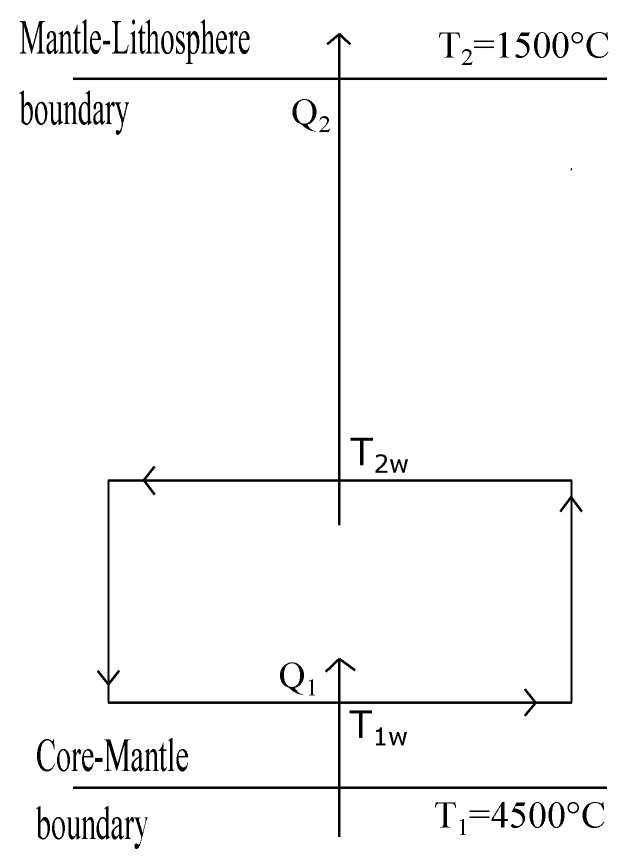
Schematic diagram of the energy fluxes present in the first internal convective cell. $T_{1}$ = 4500 ${}^{{}^{\circ}}$C is taken as the temperature of the first isothermal layer ($T_{2}$ = 1500 ${}^{{}^{\circ}}$C is taken as the cold reservoir temperature, and $T_{1w}$ and $T_{2w}$ are the internal temperatures for this endoreversible model of convective cells.

**Figure 5 entropy-20-00985-f005:**
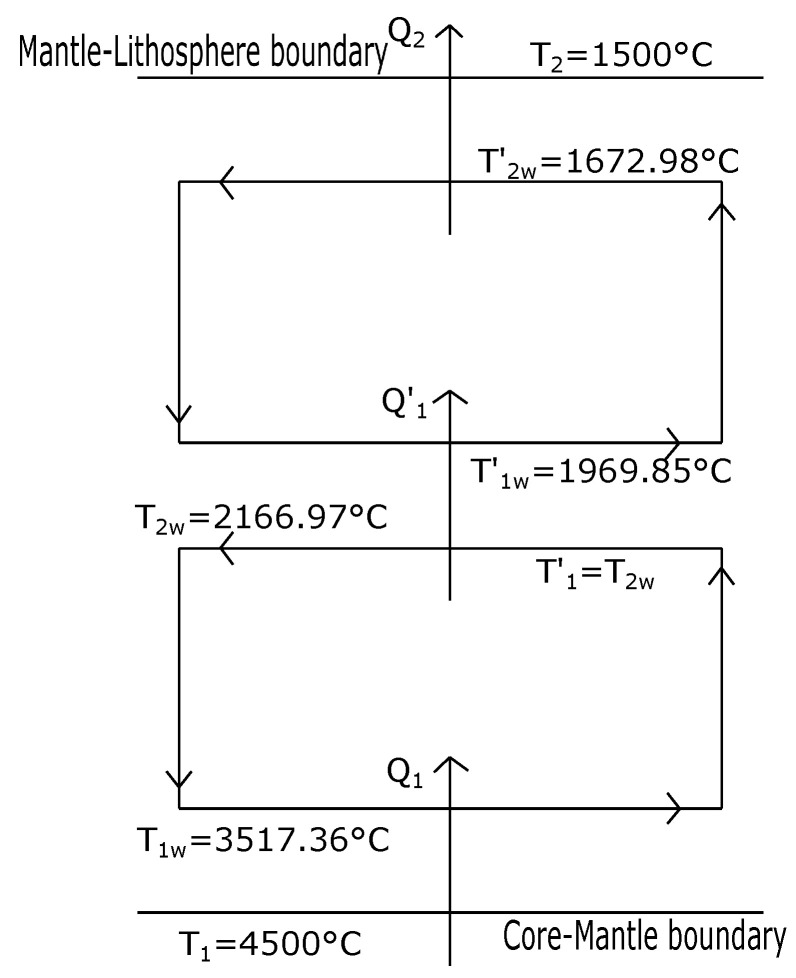
Diagram of the first and second convective layers of the Earth’s mantle for the case of maximum Power. The temperature intervals $T_{1}-T_{1w}$, $T_{2w}-T_{1w}^{\prime }$ and $T_{2w}^{\prime }-T_{2}$ approximately correspond to the D-layer, the Repetti transition zone and the 410 km - discontinuity, respectively.

**Figure 6 entropy-20-00985-f006:**
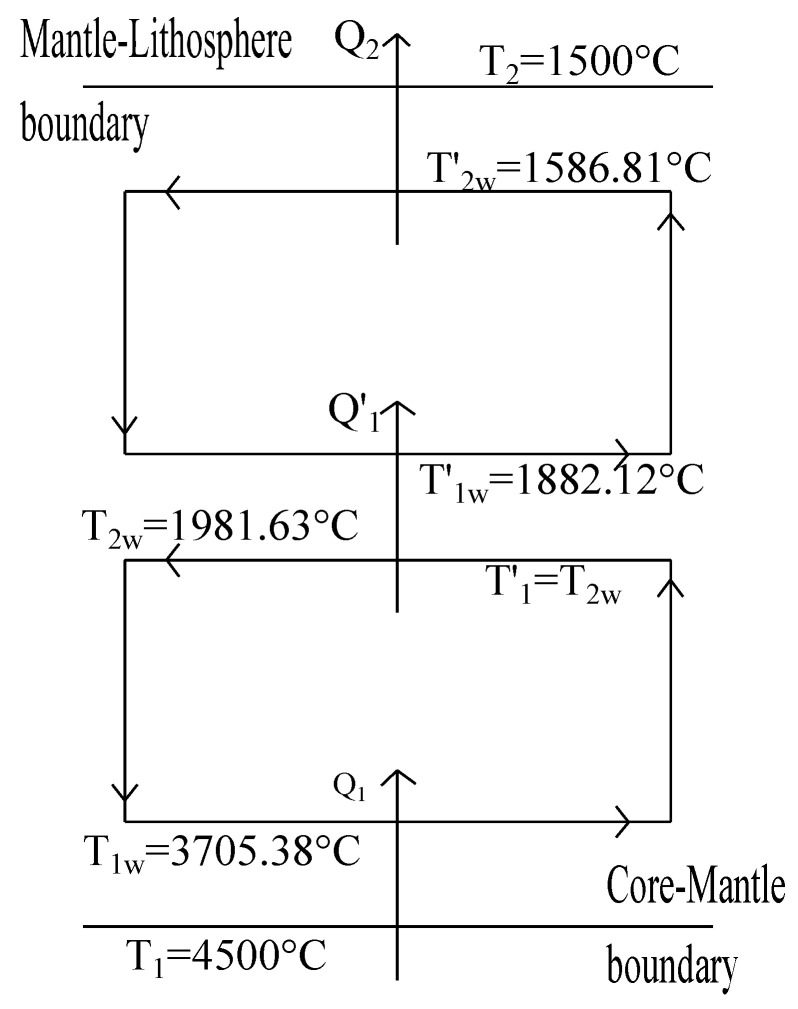
Diagram of the first and second convective layers of the Earth’s mantle for the case of maximum ecological function. The temperature intervals $T_{1}-T_{1w}$, $T_{2w}-T_{1w}^{\prime }$ and $T_{2w}^{\prime }-T_{2}$ are also not very far from those corresponding to the discontinuities mentioned in Figure 5.

**Table 1 entropy-20-00985-t001:** Numerical results for the two convective layers between the outer core and the lower limit of the crust of the Earth at maximum power conditions.

*n*	$T_{1w}$ (${}^{{}^{\circ}}$C)	$T_{2w}$ (${}^{{}^{\circ}}$C)	$T_{1w}^{\prime }$ (${}^{{}^{\circ}}$C)	$T_{2w}^{\prime }$ (${}^{{}^{\circ}}$C)
	(1st It)	(1st It)	(2nd It)	(2nd It)
1	3249.04	2049.04	1901.1	1626.58
1.2	3524.01	2145.41	1957.67	1663.95
1.25	3517.36	2166.97	1969.85	1672.98
1.5	3483.91	2261.83	2021.24	1715.69

**Table 2 entropy-20-00985-t002:** Numerical results for the two convective layers between the outer core and the lower limit of the crust of the Earth at maximum ecological function conditions.

*n*	$T_{1w}$ (${}^{{}^{\circ}}$C)	$T_{2w}$ (${}^{{}^{\circ}}$C)	$T_{1w}^{\prime }$ (${}^{{}^{\circ}}$C)	$T_{2w}^{\prime }$ (${}^{{}^{\circ}}$C)
	(1st It)	(1st It)	(2nd It)	(2nd It)
1	3256.23	2427.05	2025.06	1871.51
1.2	3709.21	1965.91	1871.34	1582.23
1.25	3705.38	1981.63	1882.12	1586.81
1.5	3685.58	2051.01	1933.14	1602.4

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
