# Peer review of "A Simple Thermodynamic Model of the Internal Convective Zone of the Earth"

_entropy, 2018, doi:10.3390/e20120985_

Round 1

Reviewer 1 Report

The authors present an FTT Gordon-Zarmi model to coarsely describe the convection in the Earth’s mantle extending a previous model for convective cells of the atmospheric air. I find the paper appropriate to be published in Entropy as the model and the obtained results accurately describe reported results of a very complicated phenomena with tools coming from FTT and two optimization criteria.

Some suggestions to improve the paper:

1.     Since sections 1 and 2 are well known results should be merged in a one section devoted to the TTF convective Gordon-Zarmi model, just making emphasis in the needed results for the new Gordon-Zarmi model of the internal convective zone of the earth.

2.     Line 214: As O’Sullivan….in his original ¿1879 paper? The cited reference 26 is from 1990.

3.     Equation 20 (as in previous Eq. 6) is not clear, at least for me. Should be improved in its presentation.

4.     Eq. 24 (as in previous Eq. 10) does not include any T(t) term; Then the condition L/T(t)=0 used to obtain Eq 25 and 26 is confusing. 

5.     In tables 1 and 2 the n-value of 1.25 applies to the Dulong-Petit coefficient. The remainder 1, 1.2, and 1.5 are different used exponents.  However, in the text (line 274) is mentioned also the n-value of 1.1. not included in the tables.

6.     In the concluding remarks, the Authors note that the results are a-independent; It is usual in FTT-models two different thermal conductances for the hot-side and cold-side heat transfers. However, here the same factor is used in the two sides according to eq. 18; may be this is the origin of the above aindependence. The authors should explain why this election. 

7.     Finally, the obtained results seem to be appropriate for both used optimization regimes. However, maximum power and maximum ecological regimes means quite different energetic interpretations. I find that this feature deserves to be commented. 

Author Response

Reviewer 1:

Comment 1. “Since sections 1 and 2 are well known results should be merged in a one section devoted to the TTF convective Gordon-Zarmi model, just making emphasis in the needed results for the new Gordon-Zarmi model of the internal convective zone of the earth.”

Answer 1. We have merged sections 2 and 3 (which are the sections 1 and 2 mentioned by the reviewer). In the corrected manuscript the fusion of these sections now is the section 2.

Comment  2: “Line 214: As O’Sullivan….in his original ¿1879 paper? The cited reference 26 is from 1990”

Answer 2. We have corrected this paragraph, which now is in the lines 216-218 as follows:

As O’Sullivan asserts [30], Stefan in his original 1879 paper, took the results of Dulong and Petit (DP) along with experiments by Tyndall and pointed out that the DP model was in agreement with his T4 law [30].”

Comment 3. “Equation 20 (as in previous Eq. 6) is not clear, at least for me. Should be improved in its presentation.”

Answer 3.  Between lines 148 and 149 we have added the following paragraph:

 “In Eq. (6) the temporal dependence of the temperatures T(t) and the heat input qs(t) is given in terms of the lower part and the upper part of the cycle period t0 respectively.”

and between the lines 235 and 236 we also have added,

“which give us the temporal dependence of the working temperatures along the two isotherms (lower and upper) of the cycle. By Using Eqs. (19) and (20) we obtain,”.

We hope that these two paragraphs improve the explanation.

Comment 4.” Eq. 24 (as in previous Eq. 10) does not include any T(t) term; Then the condition dL/dT(t)=0 used to obtain Eq 25 and 26 is confusing.”

Answer 4. In line 165 we have splitted the symbolic notation in two derivatives shown in the line just after Eq. (10), and the same for the line just after Eq. (24).

Comment 5.” In tables 1 and 2 the n-value of 1.25 applies to the Dulong-Petit coefficient. The remainder 1, 1.2, and 1.5 are different used exponents. However, in the text (line 274) is mentioned also the n-value of 1.1 not included in the tables.”

Answer 5.  In the line 280 we have deleted the exponent n = 1.1.

Comment 6.”  In the concluding remarks, the Authors note that the results are a-independent; It is usual in FTT-models two different thermal conductances for the hot-side and cold-side heat transfers. However, here the same factor is used in the two sides according to eq. 18; may be this is the origin of the above aindependence. The authors should explain why this election.”

Answer 6. We have added a new paragraph just after Eq. (19) to improve the explanation required. The new paragraph is:

 “here, in Eq. (18) we use the same coefficient of convection (a) for both the lower and upper parts of the Carnot-like cycle emulating a convective cell, under the assumption that in average the working substance (mantle materials) undergoing that convection movement has practically the same properties.”

Comment 7.” Finally, the obtained results seem to be appropriate for both used optimization regimes. However, maximum power and maximum ecological regimes means quite different energetic interpretations. I find that this feature deserves to be commented.”

Answer 7. Starting in line 290 we have added the following paragraph:

“As we said before the elapsing time of actual convective cells in the mantle is in the order of 50 - 200Myr; that is, they are very slow processes. We think that for this kind of processes both the maximum power and the maximum ecological regimes are not very different.”

Reviewer 2 Report

The paper is very interesting. Using a very simple thermodynamic model, the authors can obtain a very interesting result. I can only suggest to improve their References, Introduction and Conclusions by introducing:

- the results of Luigi Sertorio on the thermodynamics of ecosystems and complex systems

- the results of Umberto Lucia on the entropy generation

- some statements on the Constructal law approach of Adrian Bejan on the convection

- the results of Arto Annila on the approach of complex systems

After these improvement I suggest to accept the paper

Author Response

Reviewer 2:

I can only suggest to improve their References, Introduction and Conclusions by introducing:

Comment 1. “the results of Luigi Sertorio on the thermodynamics of ecosystems and complex systems”

Answer 1.  We have included a reference of L. Sertorio devoted to complex systems at the beginning of Concluding Remarks (Ref. [54])

[54] Sertorio L.; Thermodynamics of Complex Systems (An introduction to Ecophysics), World Scientific, Singapure, 1991.

Comment 2. “the results of Umberto Lucia on the entropy generation”

Answer 2. We have included a reference of U. Lucia concerning entropy generation linked to the constructal theory. The reference is the following one:

[16] Lucia U.; Exergy flows at bases of Constructal law, Physica A, 2013, 392, 6284-6287.”

Comment 3. “some statements on the Constructal law approach of Adrian Bejan on the convection”

Answer 3. We have included two references of Adrian Bejan related to the contructral theory. The references are:

 [14] Reis A. H.; Bejan A.; Constructal theory of global circulation and climate, Int. J. Heat Mass Transfer, 2006, 49, 1, 1857-1873.”

[15] Bejan A.; Shape and Structure, from Engineering to Nature, Cambridge University Press, Cambridge, 2000.”

Comment 4. “the results of Arto Annila on the approach of complex systems”

Answer 4. The approach of Arto Annila to complex systems is within the context of complex networks, which is a very important methodology to treat complex systems. However, in relation to convection in the Earth’s mantle this approach is not used yet. Due to this, we consider not pertinent a quote of this author.

Round 2

Reviewer 1 Report

all my previous comments were correctly addressed by the authors.